# Bio-Polyurethane Foams Modified with a Mixture of Bio-Polyols of Different Chemical Structures

**DOI:** 10.3390/polym13152469

**Published:** 2021-07-27

**Authors:** Aleksander Prociak, Maria Kurańska, Katarzyna Uram, Monika Wójtowicz

**Affiliations:** Faculty of Chemical Engineering and Technology, Cracow University of Technology, Warszawska 24, 31-155 Cracow, Poland; aleksander.prociak@pk.edu.pl (A.P.); katarzyna.uram@doktorant.pk.edu.pl (K.U.); monika.wojtowicz@student.pk.edu.pl (M.W.)

**Keywords:** bio-polyols, foaming process, bio-foams, physical-mechanical properties

## Abstract

We report on rigid polyurethane (PUR) foams prepared using different contents of a mixture of two bio-polyols (20–40 php). The bio-polyols were obtained through epoxidation and a ring opening reaction. Different chemical structures of the bio-polyols resulted from the use of 1-hexanol and 1,6-hexanediol as opening agents. The bio-polyols were characterized by hydroxyl values of 104 and 250 mgKOH/g and viscosities of 643 and 5128 mPa·s, respectively. Next, the influence of the bio-polyols on the foaming process of PUR systems as well as the foam properties was evaluated. The bio-foams modified with different contents of the bio-polyols were next compared with a reference foam obtained using a polyether petrochemical polyol. The effect of the apparent density reduction as a result of replacing the petrochemical polyol was minimized by decreasing the water content in the formulation. It was found that the modification of the recipe by changing the content of water, acting as a chemical foaming agent, did not affect the foaming process. However, the introduction of the bio-polyols mixture limited the reactivity of the systems by reducing the maximum temperature of the foaming process. The bio-materials with comparable apparent densities to that of the reference material were characterized by similar values of the thermal conductivity coefficient and a decrease in their mechanical strengths. A deterioration of mechanical properties was caused by the plasticization of the polyurethane matrices with the bio-polyols containing dangling chains. However, all materials were dimensionally stable at room temperature.

## 1. Introduction

Undoubtedly, the development of the chemical industry has an effect on the life of every human being. However, it is not indifferent to the state of the natural environment. The increased emission of carbon dioxide, the generation of harmful, volatile compounds that destroy the ozone layer and the pollution of surface waters have resulted in a discussion on the relationship between economic development and environmental protection. In 1992, at the The United Nations Conference on Environment and Development (UNCED), also known as the Rio de Janeiro Earth Summit, the leaders of states approved 21 principles of sustainable development as an action plan for the 21st century. The convention introduced new ideological assumptions and restrictive environmental standards, which drew the world’s attention to raw materials of plant origin and biodegradable materials.

In 2019, the global polyurethane (PUR) market was estimated to be worth $95.13 billion and is forecast to increase at an annual rate of 12% up to $149.91 billion by 2023. PUR foams have a share of 67% and dominate the PUR market [1].

PUR foams with closed-cell structures are generally rigid in nature and are applied as thermal insulation due to their low thermal conductivity, low apparent density and high strength-to-weight ratio [2]. These are the main components of the reaction mixture: polyols, catalysts, surfactants, blowing agents and isocyanate. The foaming process can be carried out by following three methods: chemical, physical and chemical/physical. Water is one of the most widely applied chemical blowing agents. In a reaction with diisocyanate, it initially generates unstable carbamic acid, which immediately decomposes into an amine and carbon dioxide Through the foaming process, it is possible to obtain porous PUR materials in a wide range of apparent densities, which makes them suitable for a variety of applications, from light open-cell foams (about 15 kg/m^3^) to microporous elastomers (1000 kg/m^3^). Singh et al. analyzed the effect of different contents of the water in the foam’s formulations on properties of PUR materials. In their experiments, the density of the PUR materials blown with water decreased from 240.1 to 56.5 kg/m^3^ as the water content increased from 0 to 3 php (parts per hundred parts of polyols) [2].

Currently, in the production of polymers, petrochemical raw materials are used primarily. However, due to environmental restrictions and the need to reduce the consumption of fossil fuels, more and more attention is paid to renewable raw materials [3,4,5,6,7,8,9,10]. An example of a new, ecological solution compliant with the principles of the so-called green chemistry is the use of vegetable or waste oils for the synthesis of polyols in the polyurethane (PUR) industry [11,12,13,14,15,16,17,18,19,20].

The primary feedstock for the bio-polyol synthesis are different natural oils, like rapeseed oil, soybean oil, sunflower oil, linseed oil as well as waste oil, such us used cooking oil. Most of these materials have to be chemically modified before application in the PUR synthesis due to the lack of hydroxyl groups [21]. Epoxidation of double bonds and oxirane rings opening is an important method to obtain bio-polyols with primary and secondary OH groups. This method makes the synthesis of bio-polyols with various chemical structures possible thanks to different ring-opening reagents. The use of monoalcohols or diols in this process leads to the formation of bio-polyols with secondary OH groups only (monoalcohols) or both primary and secondary groups (diols) [22].

The synthesis of natural epoxy oils with different contents of oxirane rings results in obtaining a wide range of hydroxyl numbers (after hydroxylation) as well as average molar masses of bio-polyols. Bio-polyols obtained through the two-step method of epoxidation and opening of oxirane rings, with hydroxyl numbers from 50 to 250 mgKOH/g and functionalities in the range of 2–6, are most often described in the literature.

Bio-polyols characterized by hydroxyl values higher than 250 mgKOH/g synthesized in epoxidation and the ring-opening reaction are characterized by high viscosities, which causes problems in spray processes of such bio-systems [23].

In a previous study, Uram et al. [24] developed a bio-polyol synthesis by epoxidation of rapeseed oil and opening of oxirane rings with 1-hexanol and 1,6-hexanediol in order to obtain bio-polyols with different functionalities and components with primary and secondary hydroxyl groups. The hydroxyl number of the bio-polyol obtained with 1.6-hexanediol was more than twice as high as that of the bio-polyol obtained with 1-hexanol and the viscosity was almost eight times higher. While using high pressure spraying devices, it is highly desirable to have a bio-polyol with a lower viscosity. A lower hydroxyl number for linear polyols means higher molecular weight and higher viscosity. For higher functionalities, the relation of hydroxyl number to the viscosity might be more complicated. Still, generally, at a similar chain structure, lower hydroxyl number means higher molecular weight and higher viscosity. The polyurethane foams obtained with the use of the bio-polyol synthesized with 1-hexanol were characterized by a much lower mechanical strength. In order to reduce the viscosity of the bio-components, it was decided to use a mixture of bio-polyols having hydroxyl numbers of 104 and 250, and viscosities of 643 and 5128 mPa·s, respectively. Moreover, it was expected that the application of bio-polyols of different functionality allows to keep both low friability good dimensional stability.

A partial replacement of petroleum-based polyols with a mixture of bio-polyols is acceptable taking into account useful properties of various polyurethane foams [5,8,14]. This study investigates the influence of mixtures of two types of bio-polyols on the foaming process, cell structures and physical-mechanical properties of PUR bio-foams

## 2. Materials and Methods

### 2.1. Foam Formulation and Preparation

Porous polyurethane bio-materials were prepared using petrochemical polyol Rokopol RF551 (oxypropylated sorbitol, PCC Rokita Brzeg Dolny, Poland). Polyol RF551 was partially replaced by a mixture of bio-polyols (P_1Hex and P_1.6Hex). The bio-polyols were synthesized using a two-step method: epoxidation and opening of oxirane rings with 1-hexanol (P_1Hex) and 1.6-heksanediol (P_1.6Hex) (Figure 1). As a catalyst of epoxidation reaction, ion exchange resin Amberlyst 15 was applied. The reaction mixture was heated to a temperature of 50 °C. Next, acetic acid was added and the reaction mixture was stirred for half an hour. After that time, hydrogen peroxide was added and the process was carried out for 6 h in a temperature of 65 °C. The molar ratios of unsaturated bonds:acetic acid:hydrogen peroxide were 1:0.35:1.40 and 1:0.24:0.96 and allowed obtaining epoxidized oils with epoxy values Ev = 0.30 and Ev = 0.21 mol/100 g, respectively. The epoxidized oils were washed with water and distilled under vacuum. In the second step, opening oxirane rings by different alcohols took place. The epoxidized oil with Ev = 0.21 mol/100 g was introduced into a three-neck flask and heated to 50 °C. Next, 1-hexanol with a solution of tetrafluoroboric acid in water (48%) was added. The amount of the catalyst was 0.45% by weight of the epoxidized oil. The reaction was carried out for 20 minutes at a temperature of 65 °C. Synthesis of P_16Hex 1,6-hexanediol and sulfuric acid as a catalyst were introduced into the reactor. The amount of the catalyst was 0.3% based on the weight of the epoxidized oil. Next, the epoxidized oil was added. The mixture was heated to 100 °C. The reaction was carried out until the epoxy value of the whole mixture was 0 mol/100 g. A detailed description of the synthesis of the bio-polyols was presented in our earlier work [24]. Characteristics of the bio-polyols and petrochemical polyol are shown in Table 1.

Polymeric methylene diphenyldiisocyanate (PMDI) with a free isocyanate groups content of 31 wt.%, was supplied by Minova Ekochem. A reactive amine catalyst providing strong blowing reaction catalysis (Polycat 218) and a surfactant (Niax silicone L-6915) were supplied by Evonik Nutriton & Care GmbH (Essen, Germany) and Momentive Performance Materials (Wilton, CT, USA). Carbon dioxide generated in the reaction of the isocyanate component with water was used as a chemical blowing agent. The PUR materials were prepared via a free-rise method according to the formulations presented in Table 2.

The polyol premix (polyol and bio-polyols, catalyst, surfactant, water) was mixed for 60s and then an appropriate amount of isocyanate was added and the whole reaction mixture was stirred for 6s. Next, the reaction mixture was poured into an open mold. The foams were conditioned for 24 h at room temperature.

### 2.2. Characterization of Foaming Process and Foam Properties

The analysis of the foaming process was performed using the foam qualification system FOAMAT (Format Messtechnik GmbH, Karlsruhe, Germany). Temperature was measured with a use of thin thermocouples. Dielectric polarization was measured using a Curing Monitor Device (CMD), which gives an insight into the electrochemical processes occurring during a foam formation. Dielectric polarization reflects the conversion degree of functional groups during a polyurethane formation. Dielectric polarization is caused by the presence of chemical compounds with a high dipole moments in raw materials used for the PUR foam preparation. As a result of the chain formation and cross-linking reaction, the mobility of dipoles is suppressed, causing a decrease in the value of dielectric polarization. A scanning electron microscope (HITACHI TM3000, Tokyo, Japan) and an optical microscope (PZO, Warsaw, Poland) revealed the morphology of cells. The analysis of the cell structure was done using the software ImageJ (version 1.53f, U.S. National Institutes of Health, Bethesda, MD, USA). The cell density was determined according to equation: N_F_ = (n/A)^3/2^, where N is s the number of bubbles in the micrograph of area A in cm^2^. The apparent density of the samples was measured as the ratio of their masses and volumes according to the ISO845 standard. The closed-cell content was determined according to the ISO4590 standard. The thermal conductivity was measured using a Laser Comp Heat Flow Instrument Fox 200. The measurements were taken at an average temperature of 10 °C (the cold plate temperature 0 °C and the warm plate temperature 20 °C) according to the ISO8301 standard. The friability was determined in accordance with ASTM C421-95. The compressive strength was investigated parallel (pa) and perpendicular (pe) to the foam rise direction according to PN-EN 826.

## 3. Results and Discussion

A modification of PUR systems with bio-polyols has a significant effect on the properties of the porous materials. It is caused, among other things, by the influence of the bio-components on the foaming process, which is an important stage in the synthesis of PUR foams. It is known that bio-polyols often decrease the PUR system reactivity. In our previous work, we reported an influence of the bio-polyols functionality on the foaming process. In the current work, the PUR system was modified with bio-polyols P_1Hex and P_1.6Hex. The system modified with bio-polyol P_1.6Hex was characterized by a higher reactivity compared to that of the corresponding system containing bio-polyol P_1Hex. The use of a mixture of bio-polyols was intended to increase the reactivity while maintaining the advantageous viscosity of the system. Bio-polyol P_1Hex was characterized by a much lower viscosity (643 mPa·s) than bio-polyol P_1.6Hex (5128 mPa·s). Figure 2 shows the influence of the bio-polyols mixture on the dielectric polarizations and temperatures of the PUR systems.

The use of the mixture of the bio-polyols caused a decrease of the reactivity of the modified systems, which was confirmed by a lower maximum temperature during the foaming process. The maximum temperature of the system in which 40 wt.% of the petrochemical polyol had been replaced with the mixture of the bio-polyols was 160 °C. For a suitable system modified only with bio-polyol P_1Hex, the temperature was 159 °C [24]. Regardless of the content of the bio-polyol mixture, no differences in the changes of the dielectric polarizations were observed. The dielectric polarizations decreased as the reactions progressed [13,25].

On the basis of the foaming process and the cellular structure of the foams, a decision was made that the systems in which maximum 40 wt.% of petrochemical polyol had been replaced with the mixture of the bio-polyols would be used in further research. The systems containing 20 and 40 wt.% the bio-polyol mixtures were modified to obtain comparable apparent densities. In the PU_20m and PU_40m systems, the water content was reduced and an analysis of the foaming process was also carried out (Figure 3).

It was found that decreasing the water content did not affect the reactivity of the PUR system, although a lower water content resulted in a lower amount of carbon dioxide released. The maximum temperature during the foaming process of the systems with a reduced water content was comparable to those of the PU_20 and PU_40 systems. The modification of the reference system with the mixture of the bio-polyols generally resulted in a reduction of the foam cell sizes. This effect also observed in studies where other bio-polyols were used. Leszczyńska et al. was observed that a modification of a PUR formulation with 30 wt.% of a rapeseed oil-based polyol obtained in the reaction of epoxidation and ring opening with diethylene glycol resulted in a significant decrease of the average pore size [26]. Due to the presence of long hydrocarbon chains, bio-polyols can act as surfactants and reduce cell sizes. The influence of the bio-polyols mixture on the cellular structures of the foams observed in our experiment is shown in Figure 4 and Table 3.

In the studies presented here, a significant effect of cell size reduction after adding the mixture of the bio-polyols compared to the PU_0 foam was observed for the materials with an apparent density comparable to that of the reference material (PU_20m and PU_40m foams). The higher the content of the bio-polyol mixture, the higher the cell density in the modified foams (Table 3).

From the point of view of the use of rigid PUR foams as thermal insulation materials, the thermal conductivity coefficient is an important property. Rigid PUR foams present lower thermal conductivity (about 0.025 W/m·K) than other thermal insulation materials that are commonly used, such as mineral wool (0.037–0.055 W/m·K), expanded polystyrene (0.03–0.04 W/m·K), extruded polystyrene (0.034–0.044 W/m·K), cellulose (0.04–0.065 W/m·K) and cork (0.04–0.05 W/m·K) [27]. Thermal conductivity is closely correlated with the content of closed cells as well as the apparent density. Figure 5 and Figure 6 show the effect of modification with the mixture of the bio-polyols on the content of closed cells, thermal insulation properties, and apparent density of the tested foams in our studies.

It was found that the introduction of the bio-polyols had no significant effect on the content of closed cells. There was a slight tendency for the closed cell content to decrease with an increasing bio-polyol content, however, these changes were within the standard deviation. It was observed that increasing the apparent density in the case of PU_20m and PU40m reduced the thermal conductivity coefficient.

The modification of the porous PUR materials with the bio-polyols reduced their apparent densities. The foam densities for samples PU_20 and PU_40 with different contents of the bio-polyols mixture are in agreement with the literature data for foams containing bio-polyols. Arniza et al. [28] concluded that increasing the amount of a bio-polyol in a PUR formulation from 30 to 50 wt.%, reduced the apparent density from 94.7 to 78.3 kg/m^3^. In order to obtain apparent densities comparable to that of the reference foams, the PU_20 and PU_40 systems were modified by reducing the water content (PU_20m and PU_40m). The compressive strengths of the PUR materials were measured in two directions given the anisotropic cell structure of the foams. The results show that increasing the content of the bio-polyols mixture led to a decrease of the compressive strengths and this an effect of decreasing the apparent densities and crosslinking densities due to the application of the bio-polyols with lower functionalities compared to that of the petrochemical polyol [23]. The modification of the recipe by reducing the water content partially reduced the effect of lower density and the materials PU_20m and PU_40m were characterized by more favorable values of compressive strength. In the case of PU_20m, the compressive strength values obtained were similar to those of the PU_0 material. The PU_40m material, despite the apparent density comparable to the that of the reference material, exhibited a worse compressive strength and this effect is related to the plasticization of the PUR matrix as a result of the introduction of the bio-polyol characterized by long dangling hydrocarbon chains acting as a plasticizer. The loss of mass during the friability test was on a satisfactory level (Figure 7). The friability of all the tested PUR materials was in the range 3–7%.

In our earlier work, the replacement of a petrochemical polyol with rapeseed oil bio-polyols increased the friability of the bio-foams. The loss of mass during the test was dependent on the type of bio-polyol. The highest friability of about 28% was obtained with the bio-polyol synthesized in the transesterification reaction, while in the case of the bio-polyols obtained in the epoxidation and oxirane ring opening reactions the friability was 13% [8].

## 4. Conclusions

The formulations of rigid polyurethane bio-foams were developed according to new trends in the synthesis of bio-materials based on renewable raw materials. The effect of apparent density reduction as a result of replacing a petrochemical polyol was minimized by reducing the water content in the formulations. The modifications of the formulations resulted in obtaining bio-foams with apparent densities comparable to that of the reference material. An increase in the cellular density was found for the foams of comparable apparent densities and modified with the bio-polyols. Such an effect was not observed in the foams in which the water content in the recipe had not been corrected. The foams with comparable apparent densities were characterized by comparable values of the coefficient of thermal conductivity and a slight reduction in compressive strengths with respect to the reference material. However, regardless of the lower compressive strengths, all the materials were dimensionally stable at room temperature. The results clearly suggest that a proper selection of the content of renewable raw materials as well as a chemical blowing agent in the formulations of rigid PUR foams allows for preparation of environmentally-friendly materials with properties similar to those of reference materials prepared using only petrochemical substrates.

## Figures and Tables

**Figure 1 polymers-13-02469-f001:**
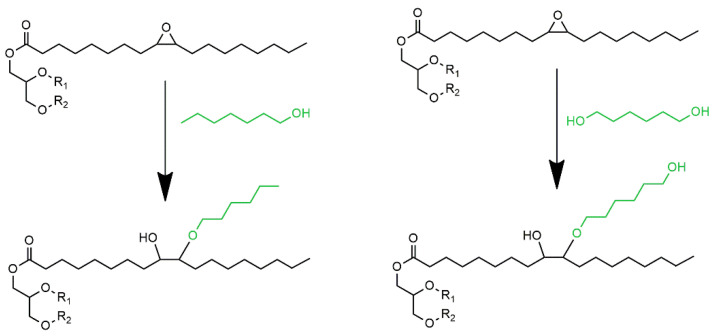
Oxirane ring opening reaction with 1-hexanol and 1,6-dexanediol.

**Figure 2 polymers-13-02469-f002:**
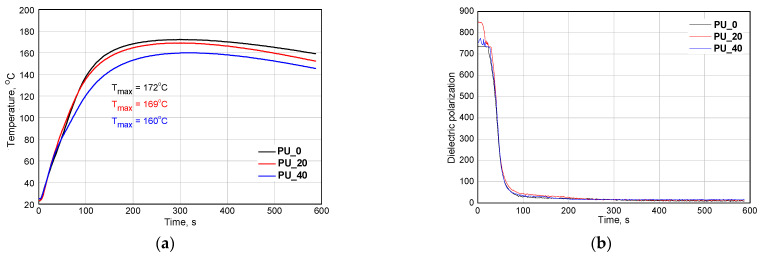
Influence of bio-polyols mixture on the changes of temperature (**a**) and dielectric polarization (**b**) of PUR systems during foaming process.

**Figure 3 polymers-13-02469-f003:**
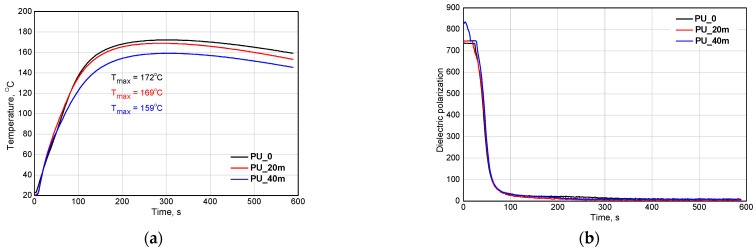
Influence of bio-polyols mixture on temperatures (**a**) and dielectric polarizations (**b**) of PUR systems with lower contents of water in formulations.

**Figure 4 polymers-13-02469-f004:**
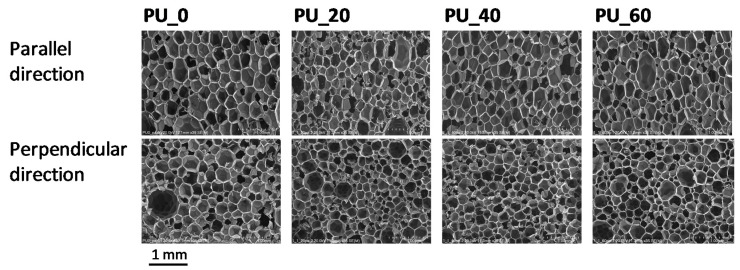
SEM microphotographs of foams.

**Figure 5 polymers-13-02469-f005:**
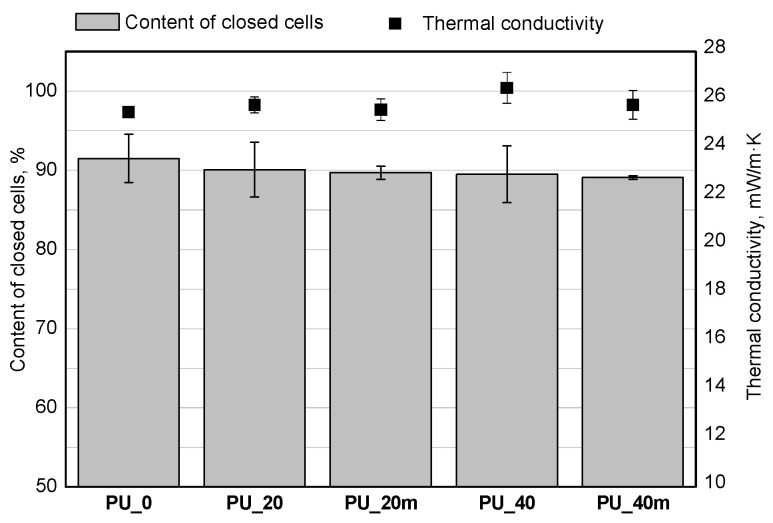
Effect of bio-polyols mixture and water content on closed cells content and thermal conductivity of foams.

**Figure 6 polymers-13-02469-f006:**
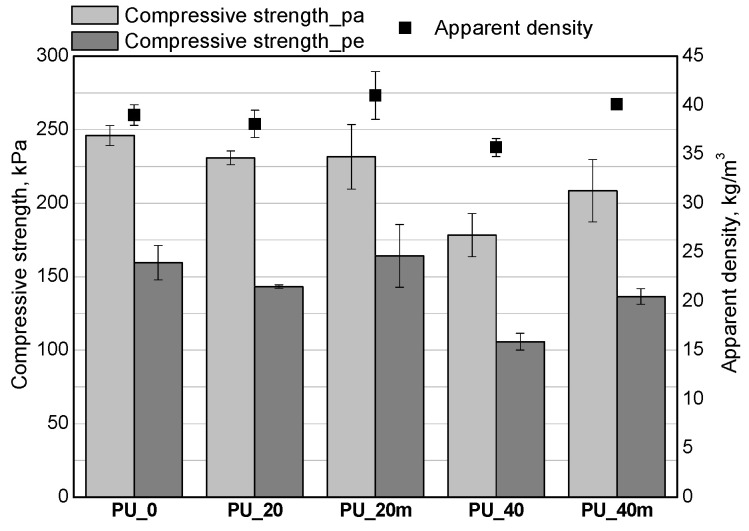
Effect of bio-polyols mixture and water content on compressive strength and apparent density of foams.

**Figure 7 polymers-13-02469-f007:**
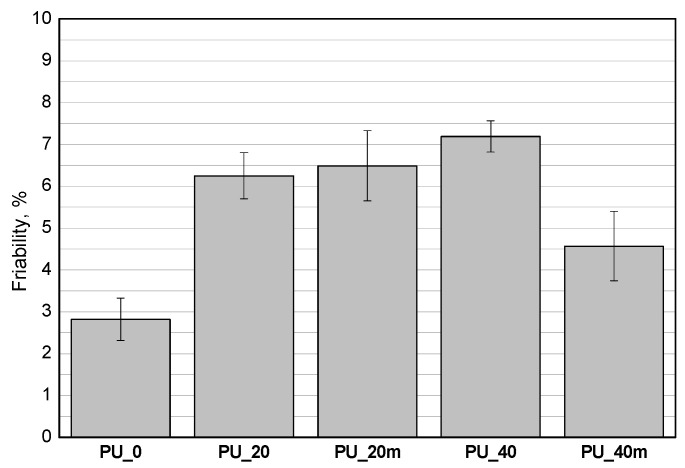
Effect of content of the bio-polyols mixture and water on friability of rigid PUR foams.

**Table 1 polymers-13-02469-t001:** Characteristics of the bio-polyols and petrochemical polyol.

Properties	P_1Hex	P_1.6Hex	RF551
Hydroxyl value, mgKOH/g	104	250	428
Acid value, mgKOH/g	1.38	4.05	0.10
Water content, %mas.	0.05	0.38	0.10
Average molecular weight, g/mol	1442	978	625
Viscosity, mPa·s	643	5128	4000
Functionality	2.67	4.36	4.77

**Table 2 polymers-13-02469-t002:** Formulations of foams.

Raw Materials, g	PU_0	PU_20	PU_20m	PU_40	PU_40m
RF551	36.00	30.40	30.40	24.00	24.00
P_1Hex	0	3.80	3.80	8.00	8.00
P_1.6Hex	0	3.80	3.80	8.00	8.00
Polycat 218	0.54	0.57	0.57	0.60	0.60
L-6915	0.54				
Water	1.22	1.22	1.20	1.22	1.18
Isocyanate index	1.1				

**Table 3 polymers-13-02469-t003:** Characteristics of PUR cellular structures.

	Symbol	Cross Section Area, mm^2^	Anisotropy Index	Cell Density
Parallel direction	PU_0	0.0063 ± 0.0006	1.12 ± 0.06	670.33 ± 82.43
	PU_20	0.0059 ± 0.0004	1.11 ± 0.02	690.79 ± 183.80
	PU_20m	0.0054 ± 0.0004	1.12 ± 0.03	710.25 ± 65.14
	PU_40	0.0068 ± 0.0009	1.17 ± 0.04	575.19 ± 51.33
	PU_40m	0.0054 ± 0.0007	1.12 ± 0.04	805.07 ± 84.30
Perpendicular direction	PU_0	0.0045 ± 0.0007	0.90 ± 0.04	948.41 ± 104.04
	PU_20	0.0045 ± 0.0004	0.99 ± 0.04	968.75 ± 77.87
	PU_20m	0.0040 ± 0.0005	0.91 ± 0.04	1029.87 ± 118.97
	PU_40	0.0047 ± 0.0011	0.93 ± 0.02	1026.74 ± 170.52
	PU_40m	0.0037 ± 0.0003	0.92 ± 0.02	1103.82 ± 56.91

## Data Availability

The data presented in this study are available on request from the corresponding author.

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
