# Peer review of "Bio-Polyurethane Foams Modified with a Mixture of Bio-Polyols of Different Chemical Structures"

_polymers, 2021, doi:10.3390/polym13152469_

Round 1

Reviewer 1 Report

Line 28:  Undoubtedly, the development of the chemical industry has a beneficial effect  the life of every human being. REMARK: Not for every human being it was or is beneficial. I suggest to cancel the word beneficial : Undoubtedly, the development of the chemical industry has an effect  on the life of every human being.

Line 44: Their main components are polyols, catalysts, surfactants, blowing agents and isocyanate. REMARK: These are the main components of the reaction mixture,  not of foams after the reaction

Line 45: The foaming process can be carried out following three methods: CORRECT: The foaming process can be carried out by following three methods

Line 52 Singh et al. analyzed the effect of different contents of water on PUR materials CORRECT Singh et al. analyzed the effect of different contents of the water in the foam’s formulations on properties of PUR materials

Line 54 (per hundred polyols) CORRECT (parts per hundred parts of polyols)

Line 56 Currently, petrochemical raw materials are mainly used in the production of polymers. CORRECT Currently, in the production of polymers, petrochemical raw materials are used primarily.

Line 62 The primary feedstock for the bio-polyol synthesis is different natural oils CORRECT are different natural oils

Line 87 However, a lower viscosity is associated with a lower hydroxyl number which results in  a lower degree of the polyurethane matrix cross-linking and, consequently, a lower mechanical strength. REMARK : A lower hydroxyl number for linear polyols means higher molecular weight and higher viscosity. For higher functionalities, the relation of hydroxyl number to the viscosity might be more complicated. Still, generally, at a similar chain structure, lower hydroxyl number means higher molecular weight and higher viscosity.

The hydroxyl number/viscosity relations of the polyols presented in Table  1 could not be compared due to differences in the topological structure of the polyols.

Line 113 prepared bvia CORRECT prepared via

Line 131:  10oC (the cold plate temperature 0oC and the warm plate temperature 20oC) REMARK : please change to upper case Celsius degree designation.

Line 132 :The brittleness was determined in accordance with ASTM C421 CORRECT The friability was determined…  REMARK : ASTM C421 - 95  Standard Test Method for Tumbling Friability of Preformed Block-Type Thermal Insulation

Line 140: bio-polyols often decrease of the PUR system reactivity CORRECT bio-polyols often decrease the PUR system reactivity

Line 148: Fig. 1 shows the influence of the bio-polyols mixture on the dielectric polarizations and temperatures of the PUR systems. REMARK: Please include more information on dielectric polarization measurements.

Line 184 Figure 3. SEM microphotographs of foams. REMARK Magnification value  is not visible.

Line 244: The loss of mass  during the brittleness test was on a satisfactory level. The brittleness of all the tested PUR  materials was in the range 3-7% . CORRECT The loss of mass during the friability test was on a satisfactory level. The friability of all the tested PUR materials was in the range 3-7% .

Line 249: Figure 6. Effect of content of the bio-polyols mixture and water brittleness of rigid PUR foams CORRECT: Figure 6. Effect of content of the bio-polyols mixture and water on friability of rigid PUR foams REMARK : the same for the y-axis : friability not brittleness

Line 252: increased the brittleness CORRECT increased the friability

REMARK please replace brittleness with friability in all other places

Line 199 and 263: cellular density REMARK: How do you define the cellular density?

Author Response

Dear Reviewer

Thank you for your quick response and very helpful comments we have received. We found them very useful and implement all of them in a manuscript. Changes included in revised manuscript have been marked with red color.

Yours sincerely                  

  1. Kurańska et al.

Line 28:  Undoubtedly, the development of the chemical industry has a beneficial effect  the life of every human being. REMARK: Not for every human being it was or is beneficial. I suggest to cancel the word beneficial : Undoubtedly, the development of the chemical industry has an effect  on the life of every human being.

It has been corrected according to Reviewer’s suggestion.

Line 44: Their main components are polyols, catalysts, surfactants, blowing agents and isocyanate. REMARK: These are the main components of the reaction mixture,  not of foams after the reaction

It has been corrected according to Reviewer’s suggestion.

Line 45: The foaming process can be carried out following three methods: CORRECT: The foaming process can be carried out by following three methods

It has been corrected according to Reviewer’s suggestion.

Line 52 Singh et al. analyzed the effect of different contents of water on PUR materials CORRECT Singh et al. analyzed the effect of different contents of the water in the foam’s formulations on properties of PUR materials

It has been corrected according to Reviewer’s suggestion.

Line 54 (per hundred polyols) CORRECT (parts per hundred parts of polyols)

It has been corrected according to Reviewer’s suggestion.

Line 56 Currently, petrochemical raw materials are mainly used in the production of polymers. CORRECT Currently, in the production of polymers, petrochemical raw materials are used primarily.

It has been corrected according to Reviewer’s suggestion.

Line 62 The primary feedstock for the bio-polyol synthesis is different natural oils CORRECT are different natural oils

It has been corrected according to Reviewer’s suggestion.

Line 87 However, a lower viscosity is associated with a lower hydroxyl number which results in  a lower degree of the polyurethane matrix cross-linking and, consequently, a lower mechanical strength. REMARK : A lower hydroxyl number for linear polyols means higher molecular weight and higher viscosity. For higher functionalities, the relation of hydroxyl number to the viscosity might be more complicated. Still, generally, at a similar chain structure, lower hydroxyl number means higher molecular weight and higher viscosity.

The hydroxyl number/viscosity relations of the polyols presented in Table  1 could not be compared due to differences in the topological structure of the polyols.

It has been corrected according to Reviewer’s suggestion.

Line 113 prepared bvia CORRECT prepared via

It has been corrected according to Reviewer’s suggestion

Line 131:  10oC (the cold plate temperature 0oC and the warm plate temperature 20oC) REMARK : please change to upper case Celsius degree designation.

It has been corrected according to Reviewer’s suggestion

Line 132 :The brittleness was determined in accordance with ASTM C421 CORRECT The friability was determined…  REMARK : ASTM C421 - 95  Standard Test Method for Tumbling Friability of Preformed Block-Type Thermal Insulation

It has been corrected according to Reviewer’s suggestion

Line 140: bio-polyols often decrease of the PUR system reactivity CORRECT bio-polyols often decrease the PUR system reactivity

It has been corrected according to Reviewer’s suggestion

Line 148: Fig. 1 shows the influence of the bio-polyols mixture on the dielectric polarizations and temperatures of the PUR systems. REMARK: Please include more information on dielectric polarization measurements.

More information concerned measurement of dielectric polarization has been added in part 2.2.

Line 184 Figure 3. SEM microphotographs of foams. REMARK Magnification value  is not visible.

 Figure 3 has been corrected.

Line 244: The loss of mass  during the brittleness test was on a satisfactory level. The brittleness of all the tested PUR  materials was in the range 3-7% . CORRECT The loss of mass during the friability test was on a satisfactory level. The friability of all the tested PUR materials was in the range 3-7% .

It has been corrected according to Reviewer’s suggestion

Line 249: Figure 6. Effect of content of the bio-polyols mixture and water brittleness of rigid PUR foams CORRECT: Figure 6. Effect of content of the bio-polyols mixture and water on friability of rigid PUR foams REMARK : the same for the y-axis : friability not brittleness

 Line 252: increased the brittleness CORRECT increased the friability

REMARK please replace brittleness with friability in all other places

It has been corrected according to Reviewer’s suggestion

Line 199 and 263: cellular density REMARK: How do you define the cellular density?

Method of calculation of cell density has been added in part 2.2

Reviewer 2 Report

                  706 / 5000 Dear authors, it's a good paper that needs some improvement:   1) It is clear that the synthesis is described in the other paper. But please add at least a short description (2 to 3 sentences) of the synthesis of the polyols. Please also add a reaction scheme. 2) The scale is missing in the microscope images (Figure 3). Please insert 3) Table 3 shows twice different values for the same direction. One direction should probably be perpendicular. 4) Please change the sentence with reference 27 (line 203). This makes little sense. 5) There are also some formatting errors: ° C and not oC   BR The reviewer

Author Response

Dear Reviewer

Thank you for your quick response and very helpful comments we have received. We found them very useful and implement all of them in a manuscript. Changes included in revised manuscript have been marked with red color.

Yours sincerely                   

  1. Kurańska et al.

Dear authors, it's a good paper that needs some improvement:  

 1) It is clear that the synthesis is described in the other paper. But please add at least a short description (2 to 3 sentences) of the synthesis of the polyols. Please also add a reaction scheme.

The short description of bio-polyol synthesis method has been added according to Reviewer’s suggestion.

2) The scale is missing in the microscope images (Figure 3). Please insert

 Figure 3 has been corrected.

3) Table 3 shows twice different values for the same direction. One direction should probably be perpendicular.

It has been corrected.

4) Please change the sentence with reference 27 (line 203). This makes little sense.

It has been corrected.

 5) There are also some formatting errors: ° C and not oC  

It has been corrected.

Reviewer 3 Report

The article analysis polyurethane foams obtained using partially replaced petroleum-based polyl with bio-polyols. The article would be interesting, however, the novelty is poorly highlighted, some results are missing. At a current state, the article is too weak to consider for publication. My comments are presented bellow:

1) It is suggested to proofread the article. There are missing some words, too big spaces between words and there are no subscripts of numbers in, e.g. apparent density dimensions, etc.

2) Authors state that a partial replacement of petroleum-based polyols with renewable ones is widely acceptable. However, there is no literature review on the results or achievements obtained by other authors regarding the mentioned case. Please make more thorough literature review and I do suggest improving the Introduction section.

3) In abstract part, authors state that all foams were stable at room temperature, however, there are no measurements. Also, it is proved that dimensional instabilities in some cased occur at higher temperature and humidity conditions. Therefore, I suggest additional dimensional stability and water absorption tests.

4) Also, he replacement of polyols may affect the foaming characteristic times. I suggest improving this article with additional measurements.

5) Water was used as a blowing agent in polyurethane formulations. therefore, it is of great interest at which moment thermal conductivity measurements were done. There are some nice articles where it is shown that water blown closed cell foams have increasing thermal conductivity value after 1, 7, 14, 30 and even more days. Have authors considered applying ageing procedure to analyse the changes in the mentioned parameter?

In my opinion, the article is not suitable for Polymers journal at the current state. Therefore, I suggest rejecting it.

Author Response

Dear Reviewer

Thank you for your quick response. Changes included in revised manuscript have been marked with red color.

Yours sincerely                  

  1. Kurańska et al.

The article analysis polyurethane foams obtained using partially replaced petroleum-based polyl with bio-polyols. The article would be interesting, however, the novelty is poorly highlighted, some results are missing. At a current state, the article is too weak to consider for publication. My comments are presented bellow:

1) It is suggested to proofread the article. There are missing some words, too big spaces between words and there are no subscripts of numbers in, e.g. apparent density dimensions, etc.

It has been corrected.

2) Authors state that a partial replacement of petroleum-based polyols with renewable ones is widely acceptable. However, there is no literature review on the results or achievements obtained by other authors regarding the mentioned case. Please make more thorough literature review and I do suggest improving the Introduction section.

This part has been completed.

3) In abstract part, authors state that all foams were stable at room temperature, however, there are no measurements. Also, it is proved that dimensional instabilities in some cased occur at higher temperature and humidity conditions. Therefore, I suggest additional dimensional stability and water absorption tests.

Good dimensional stability of such type of foams at different conditions has been confirmed in other our papers that will be published shortly.

4) Also, he replacement of polyols may affect the foaming characteristic times. I suggest improving this article with additional measurements.

Additional measurements of characteristic foaming times are not necessary. Such times are calculated from Foamat measurements and an information has been added in the manuscript.

5) Water was used as a blowing agent in polyurethane formulations. therefore, it is of great interest at which moment thermal conductivity measurements were done. There are some nice articles where it is shown that water blown closed cell foams have increasing thermal conductivity value after 1, 7, 14, 30 and even more days. Have authors considered applying ageing procedure to analyse the changes in the mentioned parameter?

Such type analysis was not considered in this paper.

In my opinion, the article is not suitable for Polymers journal at the current state. Therefore, I suggest rejecting it.

Reviewer 4 Report

Authors have corrected the article according to my remarks.

Author Response

Thank you.

Reviewer 5 Report

The present paper deals of the application of bio-polyols in the foaming process of PUR and the argument is very interesting due to the current interest for sustainable production. Despite this, from the paper it is not evident the novelty introduced from authors. Does the novelty reside in the use of bio-polyols or in the opening reaction for obtaining them? Could you explain better this aspect?

Line 9, line 55: why did you decided to use “php” that is not a common notation instead of %?

Line 22: Is “that effect” related to both thermal conductivity and mechanical strength? If yes, you must correct in “these effects”. Alternatively, please clarify the sentence.

Line 24: I suggest “All materials” instead of “all the materials”.

Dots miss in some lines (44, 171, 220).

Line 142: I suggest to add to the sentence that the FOAMAT is a Foam qualification system for clarity.

Section 2.2: how many samples were produced and tested?

Fig. 2: Is the temperature of the graph a) measured during foaming?

Line 205: remove “was”

Lines 219-220: the sentence is incomplete.

Line 203: remove “in our studies”.

Lines 253-53 please verify the sentence and the part “this an effect”. Maybe “is” is missed.

Author Response

Dear Reviewer

Thank you for your quick response. Changes included in revised manuscript have been marked with red color.

Yours sincerely                      

  1. Kurańska et al.

The present paper deals of the application of bio-polyols in the foaming process of PUR and the argument is very interesting due to the current interest for sustainable production. Despite this, from the paper it is not evident the novelty introduced from authors. Does the novelty reside in the use of bio-polyols or in the opening reaction for obtaining them? Could you explain better this aspect?

This part has been completed.

 Line 9, line 55: why did you decided to use “php” that is not a common notation instead of %?

This notation is commonly used in polyurethane technology.

Line 22: Is “that effect” related to both thermal conductivity and mechanical strength? If yes, you must correct in “these effects”. Alternatively, please clarify the sentence.

This sentence has been corrected according to Rewiever’s suggestion.

Line 24: I suggest “All materials” instead of “all the materials”.

This sentence has been corrected according to Rewiever’s suggestion.

Dots miss in some lines (44, 171, 220).

It has been corrected.

Line 142: I suggest to add to the sentence that the FOAMAT is a Foam qualification system for clarity.

This sentence has been corrected according to Rewiever’s suggestion.

Section 2.2: how many samples were produced and tested?

At least 3 samples were prepared in the research. In case of differences in the results, further ones were prepared.

Fig. 2: Is the temperature of the graph a) measured during foaming?

It has been corrected.

Line 205: remove “was”

It has been corrected.

Lines 219-220: the sentence is incomplete.

It has been corrected.

Line 203: remove “in our studies”.

It has been corrected.

Lines 253-53 please verify the sentence and the part “this an effect”. Maybe “is” is missed.

It has been corrected.

Round 2

Reviewer 2 Report

It si okay

Author Response

Thank you.